# Learning Task Representations from In-Context Learning

**Baturay Saglam** [1]   **Zhuoran Yang** [1]   **Dionysis Kalogerias** [1]   **Amin Karbasi** [1]

## Abstract

Large language models (LLMs) excel in in-context learning (ICL), adapting to new tasks via example-based prompts without parameter updates. Despite their capabilities, the internal representation and generalization of ICL tasks remain elusive. We introduce a method that encodes task information in ICL prompts by computing a single vector embedding as a weighted sum of the transformer's attention heads, optimized via gradient descent to address performance challenges. Our results indicate that current methods fail to generalize numeric tasks beyond trained lengths, exhibiting significant degradation with even minimal exceedance. Our approach not only addresses these shortcomings but also enhances performance across numeric and linguistic tasks, maintaining high task fidelity. This demonstrates our method's efficacy in deriving task-specific information from in-context demonstrations, suggesting broader applications for LLMs in ICL.

## 1. Introduction

Large language models (LLMs) based on the transformer architecture (Vaswani et al., 2017) have seen dramatic improvements in recent years. A notable feature of these models, such as GPT-3 (Brown et al., 2020), is their capability for *in-context learning* (ICL). This process involves the model receiving a prompt that includes demonstrations of a task in the form of input-output pairs. When presented with a new query input, the language model (LM) can generate the appropriate output by extrapolating from the provided examples. For instance, after being prompted with a few examples, these models are capable of producing the antonyms

of given input words. A concrete example is

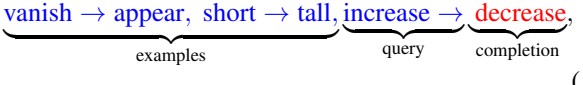

$$\tag{1}$$

where the blue text is the prompt and the red text is the completion given by an LLM.

Not limited to linguistic tasks, it has been also demonstrated that transformers can in-context learn a general class of functions $\mathcal{F}$ (Garg et al., 2023). Specifically, for any function $f \in \mathcal{F}$, the model is capable of approximating $f(x_{\text{query}})$ for a new query input $x_{\text{query}}$. This class may include linear or nonlinear relationships, potentially represented by various machine learning models such as linear regression, multilayer ReLU networks, and decision trees. This capability is particularly intriguing as it enables models to adapt to a wide range of downstream tasks on-the-fly—i.e., without requiring any parameter updates post-training (Brown et al., 2020). As the prompted in-context data points are not part of the pre-training dataset, the LM's ICL ability suggests that it can extract the task information (the relationship between the input-output pairs) from the prompt and use it to output the correct response for the query input. However, due to the complex nature of the LM's architecture, the computational mechanisms that facilitates the task encoded in the transformer's internal structure remains elusive.

Recent studies on ICL have characterized *function vectors* (FV) as a key mechanism in understanding the information flow during ICL processes (Todd et al., 2024; Hendel et al., 2023). These vectors elucidate how transformer models process and respond to various prompts. While task representation capabilities of FVs has been empirically supported, their exact computation within transformers remains a subject of debate. Early findings suggest that layer or attention activations significantly influence ICL performance (Hendel et al., 2023; Liu et al., 2024; Todd et al., 2024). However, there is no consensus on an optimal way to conceptualize or compute these vectors, leading to varied methodologies and inconsistent results across different studies.

This lack of a unified framework highlights a critical challenge in ICL research: the generalization of FVs across different task types, model architectures, and modalities. Notably, current approaches often fail to generalize to syn-

[1]Yale University. Correspondence to: Baturay Saglam <baturay.saglam@yale.edu>.

*Proceedings of the 1st Workshop on In-Context Learning at the 41st International Conference on Machine Learning*, Vienna, Austria. 2024. Copyright 2024 by the author(s).

thetic tasks[1] consisting of a range of linear and nonlinear functions. Moreover, the requirement to adapt FV computation to specific transformer models complicates the development of a standardized, automated pipeline. This issue raises concerns about the scalability and practical utility of the current formulations of function vectors. Consequently, in pursuit of a more principled conceptualization of function vectors, we hypothesize that:

*In-context learning tasks can be effectively represented by a weighted sum of all attention heads within a transformer, where the weights can be learned using gradient descent applied across the model.*

### 1.1. Contributions and Findings

Motivated by the identification of the aforementioned failure mode, our study primarily focuses on in-context learning of synthetic tasks, and also includes a brief exploration within the language domain.

**Transformers cannot generalize to longer prompts of synthetic data.** We identify a distinct failure mode in synthetic tasks, observing that when a transformer is trained with prompts containing up to $T_{\text{train}}$ examples, its ICL performance significantly deteriorates for prompts of length $T > T_{\text{train}}$. Notably, divergence consistently occurs even when $T = T_{\text{train}} + 1$. This issue, unlike the conventional generalization challenges observed in LLMs, can occur with very few tokens. For example, our experiments show that a transformer trained with $T_{\text{train}} = 41$ demonstrations (equivalent to 82 tokens) *always* fails to generalize to $T = 42$ during ICL, resulting in a significant increase in prediction error.

**An approach to assess the performance of function vectors in synthetic tasks.** If a function vector accurately represents the underlying task, it could technically extend the model's task memory beyond $T_{\text{train}}$ examples. Therefore, assessing the effectiveness of FVs on prompts longer than those used during training forms the foundation of our empirical analysis. In line with this, existing formulations of FVs either fail to enhance the transformer's performance or cannot maintain performance on prompts that exceed the length $T > T_{\text{train}}$.

**Learning weights for attention heads is superior.** We introduce a method that assigns a weight to each attention head within the multi-head self-attention mechanism of transformers, thereby computing the FV as a weighted sum of these heads' activations. The weights are optimized to enhance the

---

[1]We refer to the class of functions $\mathcal{F}$ as *synthetic tasks* since they are generated by sampling from probability distributions.

transformer's performance in scenarios where it traditionally underperforms. This approach requires learning only $L \times J$ parameters, where $L$ is the number of hidden layers and $J$ is the number of attention heads per layer. Empirical studies demonstrate that our weighted FV formulation achieves near-optimal results and effectively generalizes across longer prompts.

**Additional benefits in linguistic tasks** The proposed formulation not only excels with synthetic data but also shows advantages in linguistic tasks. Experiments reveal that our method offers a superior task representation, achieving significantly lower perplexity compared to the benchmark FV formulation (Todd et al., 2024).

## 2. Related Work

Here, we discuss studies that specifically focus on learning task representations in ICL. A more comprehensive review of related works is available in Appendix A.

Initial studies on developing task representations for transformers were documented in (Lampinen & McClelland, 2020; Shao et al., 2023; Mu et al., 2023; Panigrahi et al., 2023; Ilharco et al., 2023). These works introduced methods to create compositional task encodings through model perturbations in the parameter space, "soft" prompts, codebooks, and meta-mappings. Notably, the term *task vectors* was first used in (Ilharco et al., 2023). In contrast, by applying causal mediation analysis (Pearl, 2001; Vig et al., 2020; Wang et al., 2023a; Geva et al., 2023), function vectors were discovered to be inherently present within the transformer architecture and demonstrate strong causal effects (Todd et al., 2024). This finding parallels research in RNNs, where it was shown that RNN hidden states can be grouped based on task similarities (Lake & Baroni, 2018; Hill et al., 2019).

In this study, we aim to design a structured method for extracting function vectors from the transformer architecture, building on the approach of Todd et al. (Todd et al., 2024). Initially, Todd et al. identify a subset of attention heads, denoted by $\mathcal{A}$, across the architecture using a causal analysis metric (Pearl, 2001). An FV is then defined as the sum of activations from these heads in response to a prompt, capturing the task. This FV can enhance task performance when added to the outputs of specific hidden layers. Our approach extends this by applying learnable scalar weights to each attention head, allowing for the aggregation of all attention and the automatic adjustment of weights for heads that are less critical for task representation. Additionally, the original formulation of FVs risks disrupting the transformer's feedforward structure by potentially including attention activations from future layers into earlier layers, a point we will explore further. Despite these detrimental oversights, this formulation of FVs proves to be highly effective in language

tasks and reasonably decent, though not generalizable, in synthetic tasks. This establishes it as a benchmark in our studies.

## 3. Technical Preliminaries

The transformer architecture uses self-attention mechanisms to process sequences of data. Initially, input data is tokenized into a sequence, where each token represents data units such as words or numerical segments. In this work, we consider autoregressive transformers denoted by $M_\theta$ and parameterized $\theta$. The model predicts the next element in a sequence based on previous outputs. It consists of $L$ layers, each transforming encoded token vectors of dimension $d$ through linear and nonlinear operations. Our focus is on the computation at the last token position within these layers, where each layer $\ell \leq L$ generates a vector representation $\mathbf{h}_\ell \in \mathbb{R}^d$ from its preceding layer's output.

Self-attention in the transformer architecture employs multi-head attention at each layer:

$$\text{MultiHead}_\ell(Q_\ell, K_\ell, V_\ell) = \text{Concat}(a_{\ell,1}, \ldots, a_{\ell,j})W^O,$$

$$a_{\ell,j} \in \mathbb{R}^q := \text{Attn}_{\ell,j} = \text{softmax}\left(Q_{\ell,j}K_{\ell,j}^\top / \sqrt{d_k}\right)V_{\ell,j}.$$

Here $W^O \in \mathbb{R}^{Jq \times d}$ is the output projection matrix, and $Q_{\ell,j}$, $K_{\ell,j}$, and $V_{\ell,j}$ are the query, key, and value matrices for each attention head $j \leq J$ at layer $\ell$. The term $\sqrt{d_k}$ normalizes the softmax operation for stability, where $d_k$ is the dimension of the key matrix. This multi-head approach allows the model to dynamically adjust its focus across different parts of the input based on the context.

### 3.1. In-Context Learning

A prompt $p^t$, corresponding to task $t$, comprises a sequence of tokens including $T$ input-output exemplar pairs $\{(x_i, y_i)\}_{i=0}^T$. Here the superscript $t$ indicates the task. We note that the length of the prompts may occasionally be referred to by the number of examples they contain. Each pair demonstrates the execution of the same underlying task $t$. This set defines a functional mapping between each input $x$ and its corresponding output $y$. In addition to these exemplar pairs, each prompt includes a specific query input $x_{\text{query}}$. The goal of ICL is to use a pre-trained LLM to predict a target response $y_{\text{query}}$ corresponding to $x_{\text{query}}$, based on the prompt containing $T$ demonstrations and $x_{\text{query}}$. We consider both synthetic and language tasks.

**In-Context Learning of Synthetic Tasks**  We consider *synthetic tasks* where the transformer in-context learns a function class $\mathcal{F}$ from demonstrations. We adhere closely to the formulation proposed in (Garg et al., 2023) for training and testing the LM on synthetic data. For each prompt, a random function $f$ from $\mathcal{F}$ is sampled accord-

ing to a distribution $\mathcal{D}_\mathcal{F}$, and a set of random inputs $x_i \in \mathbb{R}^m$ for $i = 1, \ldots, T$ is drawn from $\mathcal{D}_\mathcal{X}$. These inputs are then evaluated by $f$ to produce the prompt $p^f = \{x_1, f(x_1), \ldots, x_T, f(x_T), x_{\text{query}}\}$. The output, i.e., prediction on $x_{\text{query}}$, of a pre-trained LLM is denoted by $M_\theta(p^f)$. For example, in the case of linear functions, inputs might be drawn from an isotropic Gaussian distribution $\mathcal{N}(0, I_m)$, and a random function is selected by sampling a weight vector $w$ from $\mathcal{N}(0, I_m)$, setting $f(x) = w^\top x$. Here, the task would be defined by the weight vector $w \in \mathbb{R}^m$. For nonlinear functions, possible forms of $f$ include multi-layer ReLU networks or decision trees. We employ the models pre-trained by Garg et al. (Garg et al., 2023), with the training procedure and additional details provided in Appendix B.1.2.

**In-Context Learning of Language Tasks**  In language tasks, we focus on straightforward natural language processing applications such as antonym and synonym generation, where an example is shown in (1). During the ICL inference stage, following the framework in (Todd et al., 2024), we evaluate the ICL inference abilities of a pre-trained LM rather than training it for specific tasks. We use a dataset $P^t$ that consists of in-context prompts $p_i^t$. The model $M_\theta$ processes each prompt $p_i^t$ and produces a next-token distribution $M_\theta(\cdot \mid p_i^t)$ over the words in vocabulary $\mathcal{V}$. In our empirical tests to assess robustness, we generate *corrupted* prompts $\tilde{p}_i^t$, wherein the labels within each prompt are shuffled. This shuffling breaks the direct link between the inputs $x_{i,k}$ and the outputs $\tilde{y}_{i,k}$, leaving the provided examples uninformative about the task.

## 4. Learning Task Representations

### 4.1. Motivational Observations

Our study begins with the observation that transformers, when trained with prompts containing up to $T_{\text{train}}$ examples of task $f$, effectively minimize the squared error, $(f(x_{\text{query}}) - M_\theta(x_{\text{query}} \mid p^f))^2$, during ICL inference. However, performance deteriorates when test prompts exceed $T_{\text{train}}$ examples. Even when models are initialized to accommodate $T > T_{\text{train}}$ demonstrations, they fail to maintain accuracy in predicting query inputs, resulting in significantly increased squared errors unless they are explicitly trained with $T$ examples.

If a task representation accurately encodes the task, it could sustain the model's performance beyond $T_{\text{train}}$, thereby enabling it to effectively handle longer prompt sequences during inference. We evaluated the capability of function vectors—the only previous formulation that did not diverge on synthetic data—to extend ICL to longer prompts. The results, illustrated in Figure 1, show that this approach fails to generalize across the tasks tested. Despite notable perfor-

mance in the language domain, its shortcomings in synthetic tasks raise questions about the functionality and generalizability of the existing task encoding schemes.

As our investigation deepens, we articulate the central *research questions*:

(*i*) How can we develop a task representation in a principled manner that is generalizable across different modalities, such as language or mathematical functions?

(*ii*) How can this new representation enhance performance on downstream tasks?

### 4.2. Learnable Task Vector

We introduce a framework that learns task-specific representations from in-context data, applicable to both language and synthetic domains. The proposed method *causally* optimizes the task representation, with a particular focus on scenarios where the model underperforms, such as when generalizing to longer prompts. First, we identify two primary reasons for the shortcomings of the previous work (FV). Building on these insights, we propose remedies to these drawbacks. To avoid confusion with the existing FV formulation, we have centered our method around the term *task vector*.

**Variability in contributions of attention heads** While the FV formulation sums activations across attention heads assuming equal contributions, we argue that the influence of each head varies. Some heads may significantly contribute

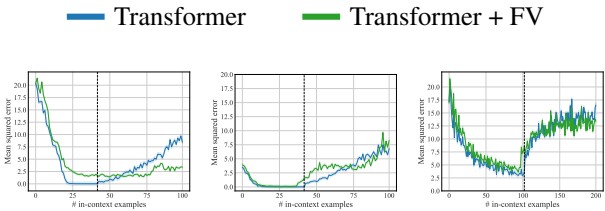

(a) Linear functions (b) Sparse linear functions (c) 2-layer ReLU NN

*Figure 1.* Squared error on the query input as a function of the number of demonstrations in the prompts. We evaluate three different function classes: (a) linear functions $f(x_{\text{query}}) = w^\top x_{\text{query}}$, (b) sparse linear functions $f(x_{\text{query}}) = w_s^\top x$, and (c) 2-layer ReLU neural networks (NNs) $f(x_{\text{query}}) = W_2 \operatorname{ReLU}(W_1 x_{\text{query}})$. Results are averaged over a batch of 256 tasks randomly selected from the same function class. The shaded area represents the 95% confidence interval over the sampled prompts. The dashed line indicates the maximum number of examples that the transformer was trained with.

to task representation, reflected by large coefficient magnitudes. Thus, constraining weights to unity is impractical, and focusing solely on a subset of attention heads, denoted by $\mathcal{A}$, may overlook subtle nuances in other heads. Hence, each attention head should be weighted in the summation to accurately represent their contribution.

**Layer-specific task vectors** A differentiated task vector, tailored to each hidden layer, would more effectively preserve the unique contributions of each layer to the task representation. Aligned with this, one might consider initializing separate sets of weights for each of the $L$ layers; however, this approach would be expensive. Instead, we propose a more efficient approach. We calculate layer-specific task vectors by computing a weighted sum of the attention heads exclusively within each layer:

$$v_\ell^t = \sum_{j=0}^{J} \omega_{\ell,j} \cdot \bar{a}_{\ell,j}, \tag{2}$$

where $\omega_{\ell,j} \in \mathbb{R}$ represents the set of weight parameters assigned to attention heads, organized in the parameter vector $\Phi \in \mathbb{R}^{LJ}$, and $\bar{a}$ represents the attention activations averaged on a separate sample set of prompts corresponding to task $t$, following (Todd et al., 2024). This method ensures that each layer-wise FV is composed solely of the $J$ heads within that specific layer, avoiding the aggregation of attention across all $L \times J$ heads.

Although excluding attention heads from layers $\ell' \neq \ell$ in the latter modification might seem contradictory, this is counterbalanced by the transformer's feedforward, autoregressive design. Including attention from earlier layers could introduce redundancy and complicate learning, as the hidden state at layer $\ell$ already encapsulates all transformed information in layers $\ell' < \ell$. Furthermore, integrating attention heads from future layers would conflict with the transformer's sequential processing, which avoids forward-looking capabilities, and could disrupt gradient flow during backpropagation, complicating training.

**How is it used during ICL inference?** Initially, a batch of sample data corresponding to a certain task $t$ is collected, and the model's attention activations are gathered. These activations are then averaged across the sample data. The average activations for each layer are weighted and summed to compute the respective layer's LTV. This task encoding is then added to the hidden states through simple vector addition to incorporate the desired task behavior or representation to the model, in alignment with the prior work (Todd et al., 2024). This process is depicted in Figure 2. We refer to the resulting refined approach as *Learnable Task Vector* (LTV) and describe the methodology for learning the weights $\Phi$ of an LTV next.

**Autoregressive transformer $M_\theta$**

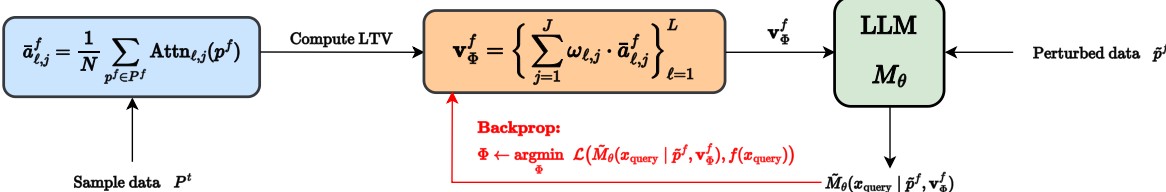

*Figure 2.* Illustration of the operation. *Additional* and *output* operations may include residual connections, normalization, feedforward, or prediction layers, depending on the architecture. Each layer's LTV is added consecutively, not simultaneously, allowing for the effects of the integrated LTV to be observed progressively across subsequent layers.

*Figure 3.* Optimization pipeline of LTV, where example is given on synthetic data. The LTV is first computed using a separate sample batch of data. It is then integrated into the model operating on perturbed data, such as longer synthetic prompts or language data with shuffled labels. The parameters are subsequently updated to minimize the loss on the query input, optimizing performance.

### 4.3. How to Optimize Learnable Task Vectors?

An effective strategy for learning the parameters $\Phi$ involves optimizing the LTV in specific scenarios where the LM typically exhibits shortcomings, while keeping the transformer parameters frozen. Hence, this approach renders our method both *data-driven* and *causal*, facilitating its application to downstream tasks with learning only $|\Phi| = L \times J$ parameters. We employ gradient descent to optimize $\Phi$ through the LM and integrate the LTV into the model for challenging/perturbed data, such as longer synthetic prompts or language data with shuffled labels. The general pipeline for training an LTV is illustrated in Figure 3.

**Synthetic Tasks**   The LTV is computed given the prompts longer than the training length $T_\mathbf{v} > T_{\text{train}}$. The integration of the LTV naturally alters the transformer's output. Subsequently, we backpropagate over $\Phi$ through the transformer to minimize the loss on the query input $x_{\text{query}}$:

$$\min_{\Phi} \mathbb{E}_{f \sim \mathcal{D}_\mathcal{F}, x \sim \mathcal{D}_\mathcal{X}} \left[ \left( \tilde{M}_\theta \left( p^f \mid \mathbf{v}_\Phi^f \right) - f(x_{\text{query}}) \right)^2 \right], \quad (3)$$

where $\mathbf{v}_\Phi^f = \{v_\ell^f\}_{\ell=1}^L$ is the set of layer-wise LTVs computed for function $f$ and $\tilde{M}_\theta(\cdot \mid \mathbf{v}_\Phi^f)$ denotes the prediction of the transformer on the query input $x_{\text{query}} \in p^f$ modified by adding $\mathbf{v}_\Phi^f$ to the corresponding hidden layers.

**Language Tasks**   A failure mode in linguistic tasks, recognized in (Todd et al., 2024), is the fragile in-context prediction performance on shuffled prompts. Hence, with the true query output $y_{\text{query}}$ known, the LTV is trained in a supervised manner to minimize the cross-entropy loss on these shuffled prompts:

$$\min_{\Phi} \mathbb{E}_{\tilde{p}^t \sim \tilde{P}^t} \left[ -\log \left( \tilde{M}_\theta \left( y_{\text{query}} \mid \tilde{p}^t; \mathbf{v}_\Phi^t \right) \right) \right], \quad (4)$$

where $\tilde{M}_\theta(y_{\text{query}} \mid \tilde{p}^t; \mathbf{v}_\Phi^t)$ is the probability predicted by the model for the true class $y_{\text{query}}$ given the shuffled prompt $\tilde{p}^t$ and when the LTV is incorporated.

## 5. Experiments

**Models**   We employ decoder-only autoregressive transformers: GPT-2 (Radford et al., 2019) for synthetic tasks and GPT-J (Wang & Komatsuzaki, 2021) for tasks in the language domain. The huggingface implementations (Wolf et al., 2020) of these models are used. GPT-2 is configured with 9.5M parameters across 12 layers and 8 attention heads per layer, while GPT-J features 6B parameters, 28 layers, and 16 attention heads per layer. The pre-trained GPT-2 models that we use adhere to the training procedure described in Appendix B.1.2 and are sourced from the

GitHub repository[2].

**Tasks**   For synthetic tasks, as outlined in Figure 1, we evaluate the models using linear functions, sparse linear functions, and 2-layer ReLU networks. Although decision trees was also explored as a function class (Garg et al., 2023), we did not encounter the same failure modes with trees; the model remains stable with increasing prompt length and can generalize effectively. For tasks in the language domain, we use the datasets curated in (Todd et al., 2024). Since our primary focus is on synthetic data, we consider two linguistic tasks: generating antonyms and synonyms.

**Function Vector**   As a benchmark for our study, we optimized the benchmark FV in synthetic tasks through several targeted modifications. These modifications significantly enhanced the effectiveness of FV, and we present the optimized configuration in our experiments with synthetic data. The details of this optimization is described in Appendix B.3. For the language domain, to ensure a fair evaluation, we directly employed the code and experimental setup outlined in the authors' GitHub repository[3].

**Learnable Task Vector**   We train LTVs for approximately 2000 iterations in synthetic tasks and 120,000 iterations in language tasks. Each iteration involves sampling a batch of input prompts with a size of 256 and performing gradient descent based on the objectives outlined in (3) and (4) for synthetic and language tasks, respectively. While we have not encountered any significant limitation in our approach since the burden of updating weights is very minimal, it is notable that creating or sampling data for LTV can consume a considerable amount of time, especially when the number of iterations is large such as in the language domain.

All details regarding our experimental setup and complete set of results are provided in Appendices B and C, respectively. To ensure reproducibility, we have made our code available in the GitHub repository (https://github.com/baturaysaglam/ICL-Task-Representations).

### 5.1. Evaluation on Synthetic Tasks

The loss curves in ICL inference are shown in Figure 4, with additional evaluation under distribution shift detailed in Appendix C.2. We note that 2-layer neural networks exhibit higher error levels than linear functions due to the complexity induced by two weight matrices and ReLU activation. FV offers some benefits in linear regression by maintaining lower error values than the vanilla transformer

for $T > T_{\text{train}}$, although the squared error of 2.5 is still high. However, FV fails to generalize to the more complex tasks and a notable error margin persists even for $T < T_{\text{train}}$. This is likely due to FVs overly perturbing the hidden states, which, while helping sustain performance beyond $T_{\text{train}}$, indicates that FVs, originally crafted for the language domain, do not effectively translate to other modalities.

**LTV yields near-optimal performance without model fine-tuning.**   LTV shows minimal performance differences when trained with prompt lengths near $T_{\text{train}}$. As $T_{\mathbf{v}}$ increases, LTV's impact grows, reaching optimal levels seen with the vanilla model for $T = T_{\text{train}}$. Notably, LTV trained with prompt lengths just above $T_{\text{train}}$—specifically, $T_{\mathbf{v}} = 1.37 \times T_{\text{train}}$ for linear functions and $T_{\mathbf{v}} = 1.25 \times T_{\text{train}}$ for 2-layer ReLU networks—suffices for maintaining this performance. Extending $T_{\mathbf{v}}$ further does not yield significant gains. This underscores that training a small subset of parameters ($|\Phi| = L \times J$) with slightly more data effectively allows the model to handle longer prompts without the need for extensive model fine-tuning.

**No special training technique is needed for LTV.**   As detailed in Appendix B.4.1, no regularization or optimization techniques, such as dropout, specific weight initialization, or specialized activation functions, are employed in learning the weights of the LTV. The weights are not bounded, nor are interventions made during the training phase. The success of this straightforward approach, yielding near-optimal results, supports the hypothesis that the ICL task may be effectively encoded as a weighted sum of the attention activations. This suggests that the weights of the attention heads naturally reach their optimal values given sufficient training.

### 5.2. Evaluation on Language Tasks

The accuracy scores are reported in Table 1. While "filtered" accuracies was primarily considered in (Todd et al., 2024), which take into account only the test queries where at least one model responds correctly, we present unfiltered accuracies as a fairer metric, counting all samples regardless of model performance. Filtered results, along with perplexity scores and losses, are available in Appendix C.3. For comparative analysis, we also trained an LTV on a joint dataset equally composed of antonyms and synonyms.

**LTV is also superior in the language domain.**   Across all tests, both LTV versions improve the vanilla transformer's performance, surpassing FV with notably higher accuracy scores and lower perplexity and cross-entropy losses. Synonym generation, more complex due to subtle semantic differences, sees LTV enhance performance from 1% to 16% in zero-shot prediction, while FV stalls at 2%. However, LTV

---

[2]https : / / github . com / dtsip / in-context-learning

[3]https://github.com/ericwtodd/function_vectors

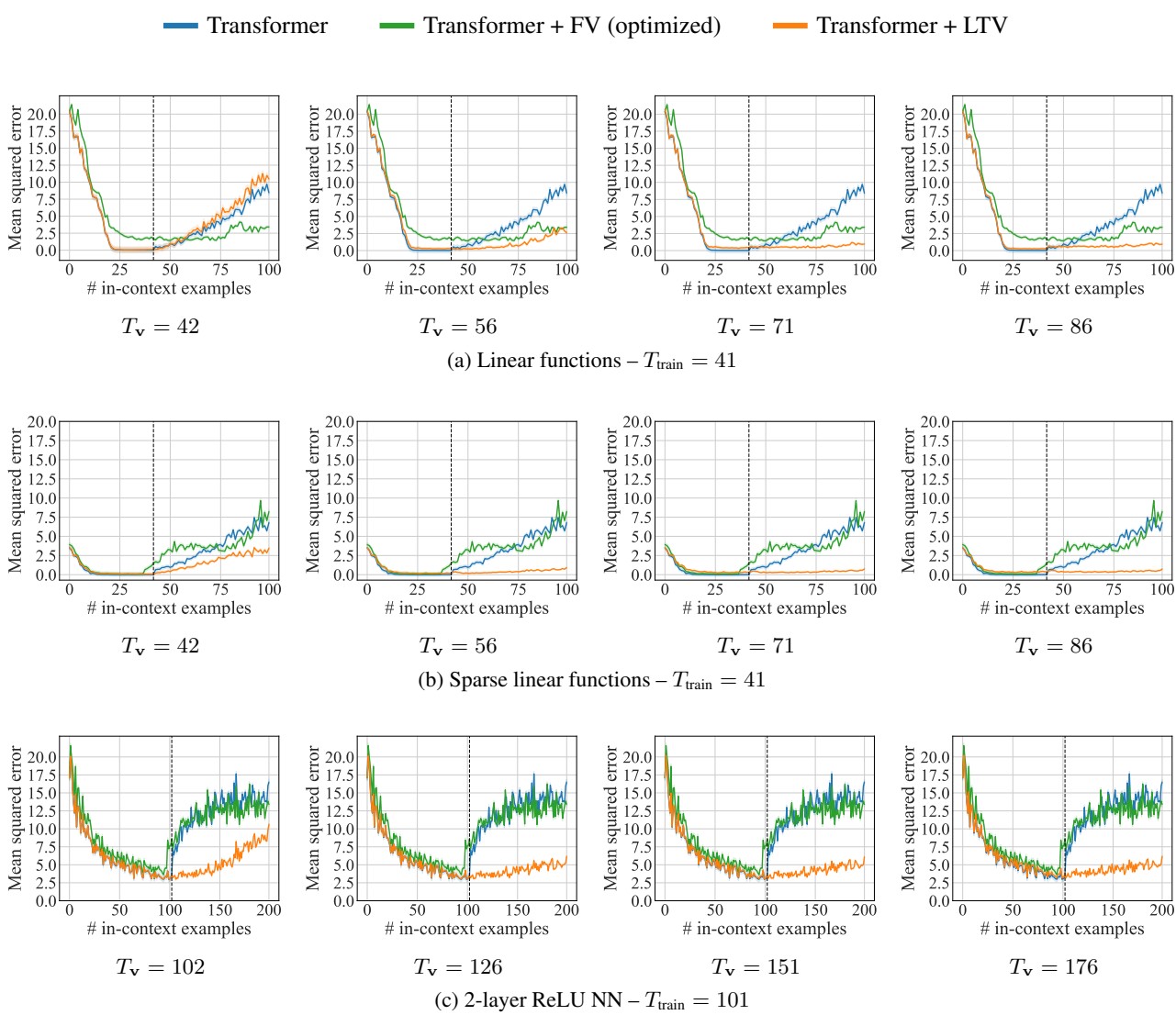

*Figure 4.* Squared error on the query input as a function of the number of demonstrations in prompts. Results are averaged over a batch of 256 tasks randomly selected from the same function class. The shaded area represents the 95% confidence interval over the sampled prompts. The dashed line indicates the number of examples the transformer was trained with, while $T_{\mathbf{v}}$ indicates the prompt length used in LTV training.

*Table 1.* Unfiltered accuracy scores for few-shot (5-shot) and zero-shot predictions, averaged across 256 random seeds. $\pm$ denotes the 95% confidence interval for the trials. The term "mixed" indicates the LTV weights trained on a joint dataset containing samples from both tasks. The highest accuracy is marked with **boldface** and highlighted.

| Model | Antonym | | Synonym | |
|---|---|---|---|---|
| | Few-shot | Zero-shot | Few-shot | Zero-shot |
| Transformer | $0.316 \pm 0.06$ | $0.023 \pm 0.02$ | $0.051 \pm 0.03$ | $0.008 \pm 0.01$ |
| Transformer + FV | $0.562 \pm 0.06$ | $0.348 \pm 0.06$ | $0.160 \pm 0.05$ | $0.023 \pm 0.02$ |
| Transformer + LTV (mixed) | $0.617 \pm 0.06$ | $0.305 \pm 0.06$ | $0.363 \pm 0.06$ | $0.062 \pm 0.03$ |
| Transformer + LTV | $\mathbf{0.641 \pm 0.06}$ | $\mathbf{0.500 \pm 0.06}$ | $\mathbf{0.402 \pm 0.06}$ | $\mathbf{0.164 \pm 0.05}$ |

*Table 2.* KL divergence values are computed between the distributions of the last hidden states of the transformer collected at $T = T_{\text{train}}$ and the listed configurations at the maximum position $T_{\text{max}}$, which is 101 for linear functions and 201 for neural networks. Kernel density estimation (KDE) is employed to estimate the probability densities using a dataset of 25,600 samples. The lowest KL divergence score (i.e., the most similar configuration) is marked with **boldface** and highlighted.

| Configuration at $T_{\text{max}}$ | Linear regression | Sparse linear regression | 2-layer ReLU NN |
|---|---|---|---|
| Transformer | 1.098 | 1.110 | 0.103 |
| + FV | 0.646 | 0.572 | 0.223 |
| + LTV – $T_{\mathbf{v}} = \{41, 41, 101\}$ | 1.000 | 1.056 | 0.095 |
| + LTV – $T_{\mathbf{v}} = \{42, 42, 102\}$ | 0.952 | 0.984 | 0.037 |
| + LTV – $T_{\mathbf{v}} = \{56, 56, 126\}$ | 0.532 | 0.260 | 0.016 |
| + LTV – $T_{\mathbf{v}} = \{71, 71, 151\}$ | 0.460 | 0.203 | 0.017 |
| + LTV – $T_{\mathbf{v}} = \{86, 86, 176\}$ | 0.344 | 0.199 | **0.013** |
| + LTV – $T_{\mathbf{v}} = \{101, 101, 201\}$ | **0.300** | **0.096** | 0.038 |

trained on a mixed dataset performs slightly worse than its task-specific counterpart, as common weights across tasks sacrifices some efficacy. These findings confirm that individual weight learning and layer-wise optimization of task vectors excel in both synthetic and challenging linguistic tasks.

### 5.3. Ablation Studies

We aim to understand how and why LTVs sustain performance beyond the training duration $T > T_{\text{train}}$. Our focus is on the last hidden states as they accumulate the most refined representations for predictions, which is crucial when the prompt length varies. We use the Kullback-Leibler (KL) divergence to measure the distributional stability of these last hidden states, assessing how closely the model's outputs with extended prompts align with those observed within the training prompt length. The results are reported in Table 2. A detailed description of our experimental methods, including a figure illustrating the pipeline, are provided in Appendix B.5. Additionally, probability densities of the last hidden state distributions, visualized using histograms, are depicted in Appendix C.4.

**LTV maintains the last hidden state distribution with that of the optimal-performing model.** We observe that the divergence between the last hidden states of the optimal-performing model (empirically, for $T = T_{\text{train}}$) and the LTV-integrated model decreases as the LTV training length increases. This observation supports the finding that extending the LTV training prompt length reduces prediction errors. Therefore, the effectiveness of the LTV can be linked to its ability to closely align the distribution of the last hidden state with that of the optimally performing model. Specifically, when the LTV training length matches or exceeds roughly the middle length (56 for linear models and 126 for 2-layer NN functions), the KL divergence reduces significantly.

**Learned LTV weights do not exhibit an interpretable pattern.** We observed that the learned attention weights, with magnitudes generally between [-3, 3], showed no consistent distribution pattern across different tasks, training durations of the LTV, or those found by Todd et al. (2024). This suggests that LTV weights might capture subtle nuances and be very sensitive to small changes in the training setup, such as the LTV training length.

## 6. Conclusion

In this study, we examine how tasks in in-context learning (ICL) are represented by large language models (LLMs). Motivated by the empirical observation that transformers do not generalize well to numerical ICL examples beyond the horizon encountered during training, we investigate whether task representations developed for language tasks can effectively address this limitation. Finding the existing representations insufficient, we propose a principled formulation that respects the feedforward nature of autoregressive transformers. This new approach represents ICL tasks through a weighted sum of attention head activations, optimized via gradient descent to enhance LLM performance in challenging scenarios.

Our findings indicate that this method not only preserves task fidelity but also enhances performance on prompts longer than those encountered during training. This improvement is achieved by training only *a few* parameters, rather than a vast array of fine-tunable transformer parameters. Consistent with these results, our proposed formulation also excels in language tasks, achieving near-optimal performance and surpassing the prior work. Ultimately, we believe this study not only opens new avenues for employing LLMs in diverse ICL applications but also establishes a promising direction for future research focused on refining and expanding the applicability of our approach.

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

# A. Extended Related Work

A substantial body of research on ICL has been ongoing since its introduction. We review previous studies through different facets of ICL. While our study shares similarities with others, it most closely aligns with the studies described in Section 2, where complementary details are provided in the final paragraph of this section.

**In-Context Learning**    The ICL capabilities of LLMs were first identified in (Brown et al., 2020). Since then, ICL has been extensively studied from various angles. The effects of different ICL prompts styles have been examined (Min et al., 2022; Yoo et al., 2022). ICL during inference time has also been explored through meta-learning analyses in references such as (Akyürek et al., 2023; Dai et al., 2023; Von Oswald et al., 2023; Li et al., 2023b; Garg et al., 2023). In addition, investigations into ICL task inference from a Bayesian perspective have been conducted (Xie et al., 2022; Wang et al., 2023b; Wies et al., 2023; Zhang et al., 2023). Additionally, the scalability of ICL across different model sizes has been examined (Wei et al., 2023; Wang et al., 2023b; Pan et al., 2023). While these studies primarily focus on the externally observable behaviors of models during inference and ICL, our study delves into the internal mechanisms of transformers to encode ICL tasks.

**The Role of Attention Mechanism in Explaining Model Behavior**    Past analyses of the attention mechanism (Voita et al., 2018; Clark et al., 2019; Voita et al., 2019; Kovaleva et al., 2019; Reif et al., 2019; Lin et al., 2019; Htut et al., 2019; Kobayashi et al., 2020) have revealed that attention weights often align with linguistic structures. However, these studies primarily focused on explaining the behavior of bidirectional architectures. Moreover, attention scores alone have not been found to fully explain the model's outputs (Jain & Wallace, 2019; Wiegreffe & Pinter, 2019; Bibal et al., 2022). In our work, we aim to deepen the understanding of the role of multi-head self-attention in ICL. Specifically, we investigate the contribution of each attention head to the model's internal representation of the ICL task, presenting interpretable findings that demonstrate how content of information is transported within the transformer architecture.

**Mechanisms to Explain Task Performance in In-Context Learning**    The components of transformers during ICL inference have been investigated to identify the origins of incorrect predictions and false statements (Merullo et al., 2024; Halawi et al., 2024). Similarly, numerous studies have adjusted attention mechanisms or activations of hidden layers during inference to steer model behavior (Li et al., 2023a; Subramani et al., 2022; Turner et al., 2023; Rimsky et al., 2024; Liu et al., 2024). It was observed that tokens representing labels in an ICL prompt might hold the semantic information crucial for the language model's final prediction (Wang et al., 2023a). Moreover, it was suggested that certain neurons within pre-trained transformers are highly predictive of task labels and empirically demonstrate that these neurons encode task-specific skills (Wang et al., 2022). In contrast to these methods, our paper seeks to develop a principled conceptualization (e.g., a function of the transformer's components) that can effectively *represent* and *differentiate* a variety of tasks across a distribution, regardless of the task modality. We further use the designed conceptualization to steer the language model's behavior towards various tasks, similar to the previous works described next.

**Tasks Representations in In-Context Learning**    It was suggested that tasks may be represented at one of the layer activations at the last token position (Hendel et al., 2023), while it was shown that the principal direction of the layer activation differences can effectively direct the ICL task (Liu et al., 2024). Nonetheless, it has been recently argued that the focus should be on attention heads (Todd et al., 2024), as these are crucial for transferring information between token positions (Vaswani et al., 2017; Elhage et al., 2021). To address these varying methodologies, function vectors are computed as the sum of the outputs from a selectively chosen subset of attention heads based on an *indirect metric* (Todd et al., 2024), derived from causal inference literature (Pearl, 2001). To the best of our knowledge and based on our preliminary analyses, the most effective empirical representation of tasks in ICL was provided in (Todd et al., 2024). Therefore, our approach starts by deriving from (Todd et al., 2024), in contrast to methods the outlined in (Hendel et al., 2023; Liu et al., 2024). Instead of merely using raw activations, we optimize weights assigned to these heads to enhance transformer performance in scenarios where it typically underperforms. Ultimately, this leads to a more formalized conceptualization that can be adapted to various models and tasks, whether synthetic or linguistic.

## B. Experimental Details

### B.1. Experiments on Synthetic Data

We precisely follow to the experimental setup established in (Garg et al., 2023). For completeness, we also provide the relevant details here. Additional information can be found in the cited reference.

#### B.1.1. MODEL

The GPT-2 model processes a sequence of vectors in the embedding space and outputs a sequence in the same space. However, the tasks we examine involve functions from a lower-dimensional vector space (e.g., 20 dimensions) to a scalar value. To use a prompt such as $p = \{x_1, f(x_1), x_2, f(x_2), \ldots, x_{\text{query}}\}$, we must map $x_i$ and $f(x_i)$ into the embedding space. This mapping involves first converting the scalar values $f(x_i)$ into vectors of the same dimension as $x_i$ by appending zeros, followed by applying a learnable linear transformation to all these vectors into the embedding space. The model's output vector is then transformed into a scalar value through a dot product with a learnable vector.

We consider the model's prediction at the position corresponding to $x_i$ (i.e., the absolute position $2i - 1$) as the prediction of $f(x_i)$. Due to the model's structure, this prediction relies solely on the pairs $(x_j, f(x_j))$ for $j < i$ and $x_i$ itself. We disregard the model predictions at positions corresponding to $f(x_i)$.

The GPT-2 models were trained to accommodate up to 101 examples for linear and sparse linear functions, and up to 201 examples for 2-layer ReLU neural networks in a prompt. While it is possible to feed the model prompts with more examples by adjusting the initialization, this would exceed our computational resources. We also did not want to alter the nature of their experimental process.

#### B.1.2. TRAINING

We train a model from scratch (i.e., no pre-trained weights are loaded) to predict $f(x_i)$ for a given $x_i$, using the set of examples as reference. Each training prompt is generated by randomly sampling a function $f$ from the function class of interest, followed by sampling inputs $x_i$ from an isotropic Gaussian distribution $N(0, I_m)$. The prompt is constructed as $p = \{x_1, f(x_1), \ldots, x_k, f(x_k)\}$. For each input $i \leq k$ within a prompt, the model predictions $\hat{y}_i = M_\theta(x_i \mid p = \{x_1, f(x_1), \ldots, x_{i-1}, f(x_{i-1})\})$ are obtained, and the loss is computed across all prompt prefixes::

$$\min_\theta \mathbb{E}_{f \sim \mathcal{D}_\mathcal{F}, x \sim \mathcal{D}_\mathcal{X}} \left[ \frac{1}{T+1} \sum_{i=0}^{T} \left( M_\theta\big(p^{f,i} = \{x_1, f(x_1), \ldots, x_i, f(x_i), x_{i+1}\}\big) - f(x_{i+1}) \right)^2 \right],$$

where $\mathcal{L}(\cdot, \cdot)$ is the loss function, typically chosen to be mean squared error, and we have $x_{T+1} = x_{\text{query}}$.

During training, we average the loss across a batch of randomly generated prompts, each with different functions and inputs, and update the model parameters. The Adam optimizer (Kingma & Ba, 2015) is employed and trained for a total of 500,000 steps with a batch size of 64, using a learning rate of $10^{-4}$ for all function classes and models.

**Curriculum Learning**  The training procedure is accelerated through curriculum learning. The model starts by observing prompt inputs $x_i$ within a smaller dimensional subspace and with fewer inputs per prompt. Both the subspace dimension and the number of examples are increased gradually. Specifically, all of the coordinates except the first $T_{\text{max, cur}}$ of $x_i$ are zeroed out by sampling prompts of size $T_{\text{cur}}$. For the function classes of linear and sparse linear functions, $T_{\text{max, cur}} = 5$ and $T_{\text{cur}} = 11$ are used initially, and $T_{\text{max, cur}}$ and $T_{\text{cur}}$ are increased by 1 and 2, respectively, every 2000 steps until reaching $T_{\text{max, cur}} = T_{\text{max}}$ and $T_{\text{cur}} = 2m + 1$. A different schedule is applied for 2-layer neural networks to accommodate the need for more inputs; starting from $T_{\text{max, cur}} = 5$ and $T_{\text{cur}} = 26$, $T_{\text{max, cur}}$ and $T_{\text{cur}}$ are incremented by 1 and 5 respectively, every 2000 steps until $T_{\text{max, cur}} = T_{\text{max}}$ and $T_{\text{cur}} = 5m + 1$.

Consequently, in the curriculum-based training approach, a training prompt $p = \{x_1, f(x_1), \ldots, x_{T_{\text{cur}}}, f(x_{T_{\text{cur}}})\}$ is generated by sampling a random function $f$ from the function class and drawing inputs $x_i$ by sampling i.i.d. from $\mathcal{N}(0, I_m)$, with all but the first $T_{\text{max, cur}}$ coordinates zeroed out. Given the model predictions $\hat{y}_i$, the loss is computed as

$$\frac{1}{T_{\text{cur}}} \sum_{i=1}^{T_{\text{cur}}} \left( \hat{y}_i - f(x_i) \right)^2.$$

### B.1.3. SAMPLING THE FUNCTIONS

For the class of linear functions, we sample the random function $f(x) = w^\top x$ by drawing $w \sim \mathcal{N}(0, I_m)$. In the case of sparse linear functions, $w$ is also sampled from $\mathcal{N}(0, I_m)$, but we then randomly zero out the first $T_{\mathrm{cur}}$ coordinates within the first $T_{\mathrm{max, cur}}$ coordinates. For these linear functions, we set $m = 20$ for all experiments, with a sparsity level of 3. For 2-layer neural networks, we sample $W_1$ from $\mathcal{N}(0, I_m)$ and $W_2$ from $\mathcal{N}(0, 2/r)$, where $f(x) = W_2 \mathrm{ReLU}(W_1 x)$. Here, we set the dimensions $m = 20$ and the ratio $r = 100$.

### B.1.4. EVALUATION

To assess performance, we sample a prompt with a maximum length of $T_{\max}$, which is equal to 101 for linear and sparse linear functions and 201 for 2-layer networks. We then trim the prompt to $T_i \leq T_{\max}$ demonstrations and independently track the prediction errors for each $i$. Consequently, each point in our error curves corresponds to the error at a specific prompt length $i$. This analysis is conducted over batches of 256 prompts, with the average error reported. We have determined that batches larger than 256 prompts do not significantly alter the results, confirming that 256 prompts are sufficient to produce generalized results.

## B.2. Experiments on Language Data

We closely adhere to the experimental methods described in (Todd et al., 2024), providing all relevant details to ensure our report is self-sufficient.

### B.2.1. DATASETS

The antonym and synonym datasets used are compiled in (Todd et al., 2024), originally based on data from (Nguyen et al., 2017), and can be accessed from their GitHub repository[3]. An initial dataset is assembled by combining all adjective, noun, and verb pairs from all data splits and removing duplicate entries. The dataset has been further refined to include only those word pairs where both words can be tokenized as a single token. As a result, this refinement retained 2398 antonym word pairs and 2881 synonym word pairs. The corresponding vocabulary size is $|\mathcal{V}| = 50,400$.

These datasets originally included multiple outputs for single inputs, e.g., "increase" $\rightarrow$ "decrease" and "increase" $\rightarrow$ "reduce". However, handling such cases would require a more powerful model (Todd et al., 2024). Therefore, the dataset has been simplified to ensure a one-to-one mapping between terms.

### B.2.2. PROMPTING

The default template for prompting the GPT-J model with $T$ exemplars is structured as follows:

$$\mathbb{Q}:\{x_1\}\backslash\mathrm{nA}:\{y_1\}\backslash\mathrm{n}\backslash\mathrm{n}\ldots\mathbb{Q}:\{x_T\}\backslash\mathrm{nA}:\{y_T\}\backslash\mathrm{n}\backslash\mathrm{nQ}:\{x_{\mathrm{query}}\}\backslash\mathrm{nA}:$$

In our experiments with shuffled prompts, we randomly shuffle the labels $\{y_i\}_{i=1}^T$ among each other. For zero-shot prompts, which contain no exemplars, prompts consists solely of the query: $\mathbb{Q}:\{x_{\mathrm{query}}\}\backslash\mathrm{nA}:$.

## B.3. Optimizing the Function Vector

Function vectors were initially proposed for language data. However, our preliminary results showed notable effects on synthetic data using the setup designed for language, and further improvements were achieved through extensive fine-tuning.

Unlike the original approach which adds the FV to only one layer, our modifications include integrating FVs into multiple layers of the GPT-2 architecture, which enhances performance. We also introduce *dummy* examples, such as (0, 0) pairs, at specific positions within the prompts to minimize output disruption while allowing the addition of FVs at these positions to "remind" the model of the task. Furthermore, our findings indicate that incorporating 35 attention heads in the FV computation significantly outperforms the typical 10 used in the original study. After tuning the scale of the FV to 1.0, we observed optimal performance. We summarize our FV optimization as follows:

1. **FV added to multiple layers:** Each added FV is divided by the number of layers it integrates into, normalizing its impact on layer activations. For GPT-2, layers 6, 7, and 8 are identified as the most effective for integrating FVs.

2. **Dummy tokens:** Positioning dummy tokens at the 0.1, 0.25, 0.5, 0.75, and 0.9 fractions of the prompt length optimizes

performance.

3. **Number of attention heads in the FV computation:** Using 35 attention heads maximizes GPT-2 performance, with worse performance beyond this number.

4. **Scale of the added FVs:** Optimal performance is achieved without scaling, aligning with findings in (Todd et al., 2024) for language tasks.

## B.4. Training the Learnable Task Vector

The LTV parameters are initially uniformly initialized between 0 and 1, totaling $L \times J$ learnable parameters. We did not employ additional techniques such as dropout, activation functions, or gradient clipping in learning the weights, which are neither clipped nor bounded. The Adam optimizer, with a learning rate of $5 \times 10^{-5}$, was used in all experiments. Training is terminated if the validation loss does not decrease for 50 consecutive gradient steps. While transformer parameters remain frozen, we still backpropagate through them.

### B.4.1. LEARNING ON SYNTHETIC DATA

We train the LTV weights using mini-batch gradient descent on a dataset we compiled, consisting of $100 \times 256 = 25,600$ function samples (100 times the batch size). Prompts of length $T_{\mathbf{v}} > T_{\text{train}}$ are constructed for these functions. We reserve the 20% of the dataset as a validation set for monitoring loss. For each mini-batch of prompts $\{p_i^{f_i}\}_{i=1}^N$ sampled from the dataset, the gradient descent step is defined as:

$$\mathcal{L}(\Phi) = \frac{1}{N} \sum_{i=1}^{N} \left( \tilde{M}_\theta \big( p_i^{f_i} \mid \mathbf{v}_\Phi^{f_i} \big) - f_i(x_{i,\text{query}}) \right)^2,$$
$$\Phi \leftarrow \Phi - \eta \cdot \nabla \mathcal{L}(\Phi),$$

where $p_i^{f_i}$ represents a prompt corresponding to a unique function $f_i$ and $\eta$ is the learning rate. This method and dataset compilation are applied uniformly across all three function classes.

We recognize that generating distinct functions and prompts at each gradient step could potentially provide an infinite variety of data and functions. However, this raises concerns about whether the LTV might memorize and overfit to specific function classes. Although it was argued that the likelihood of the model encountering a training-similar prompt is extremely low (Garg et al., 2023), we opted for a static dataset approach. Our experiments were conducted using this dataset, with evaluations performed on the prompts sampled separately during the ICL inference.

### B.4.2. LEARNING ON LANGUAGE DATA

Given the dataset $P^t$ for task $t$, each gradient step involves sampling 100 (50-50 for the mixed LTV training) prompts with replacement, each containing 5 demonstrations. These prompts are processed by the transformer to collect attention activations, and the mean of these activations across the sampled prompts is computed. This mean is then passed through the LTV layer to compute the corresponding LTV, as illustrated in (2).

For training, we sample a batch of 32 (16-16 for the mixed LTV training) prompts, each with 5 demonstrations, but with all labels shuffled, rendering the input-output pairs non-informative. The LTV weights are updated to maximize the probability of the correct label for the query input:

$$\mathcal{L}(\Phi) = -\frac{1}{N} \sum_{i=1}^{N} \log \left( \tilde{M}_\theta \big( y_{\text{query}} \mid \tilde{p}_i^t; \mathbf{v}_\Phi^t \big) \right).$$

## B.5. Detailed Ablation Studies

Our argument is based on the premise that while earlier layers build foundational representations, the most refined and actionable insights for predictions are concentrated in the outputs of the last layer. The motivation for these studies is to examine the model's resilience to variations in prompt length.

To this end, we freeze all transformer and LTV parameters to analyze the stability of last hidden state distributions across varying prompt lengths, rendering stationary distribution. We generate a dataset of 25,600 prompts (100 times the batch size) with a maximum length $T_{\max}$. Prompts were trimmed to $T_{\text{train}}$ for the vanilla transformer and extended beyond $T_{\text{train}}$ for the transformer integrated with FV and LTV. The last hidden states from these configurations were compiled into two datasets, $X_1$ and $X_2$, for the vanilla transformer and FV- and LTV-integrated transformers, respectively. Thus, the samples within each dataset are independent and identically distributed. Consequently, it is necessary to estimate the probability distributions from these samples.

We estimate probability distributions using KDE, where the standard KDE implementation from `SciPy` (Virtanen et al., 2020) is used with default settings. However, directly estimating a probability distribution in a high-dimensional space often leads to the *curse of dimensionality*, where the volume of data required to effectively estimate the distribution grows exponentially with the number of dimensions. A practical solution to this challenge is to employ SVD for dimensionality reduction. This involves decomposing the data matrices $X_i \in \mathbb{R}^{M \times d}$ as:

$$X_1 = U_1 \Sigma_1 V_1^\top,$$
$$X_2 = U_2 \Sigma_2 V_2^\top,$$

where $M$ is the number of collected samples and $U_i$ contains the principal components $u_{i,1}, u_{i,2}, \ldots, u_{i,n}$ as column vectors. These principal components (PCs) form the column space of $X_i$:

$$\text{span}(u_{i,1}, u_{i,2}, \ldots, u_{i,n}) = \text{colspace}(X_i),$$

where each $u_{i,k}$ is orthogonal to $u_{i,k' \neq k}$ and ordered by decreasing variance that they explain. Specifically, the first $n$ principal components represent the directions along which the data varies the most, capturing the most significant patterns in the data. These components are likely more informative and relevant for distinguishing different behaviors or properties of the data.

Rather than estimating the distribution underlying the entire datasets $X_i$ as multivariate distributions, we employ Gaussian KDE[4] to estimate each PC as a unimodal distribution. This approach is advantageous since KDE performs better with univariate data. However, transitioning from multivariate to univariate requires the assumption that the PCs are uncorrelated. We validate this assumption by observing that the nondiagonal entries of the correlation matrices of $U_i$ are on the order of $10^{-2}$, with diagonal entries being approximately 1, effectively an identity function, confirming that the PCs are indeed uncorrelated.

We use KL divergence between the KDE-estimated distributions of the column pairs to quantitatively assess distributional similarities. However, we find that these divergence values are negligibly small, except for the first one, which accounts for the most variance within the dataset. The negligible divergence scores for higher-order components suggest that these vectors contribute minimally to differentiating the datasets. Thus, focusing on the first component, which shows substantial divergence, is statistically justified and highlights the critical variations relevant to model generalization.

This experimentation process is depicted in Figure 5. Additionally, histograms illustrating the KDE-estimated distributions of the first principal components are provided in Appendix C.4 to offer a clear view of their similarity.

---

[4] https://docs.scipy.org/doc/scipy/reference/generated/scipy.stats.gaussian_kde.html

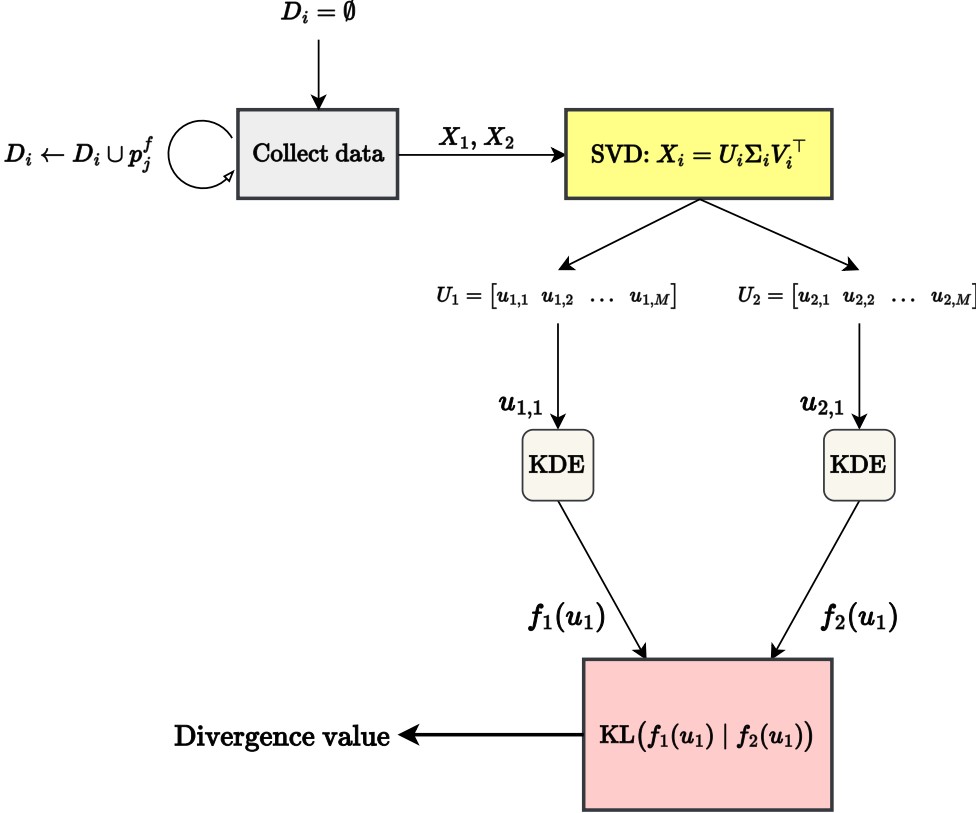

*Figure 5.* The diagram depicts our pipeline for ablation studies. We begin by collecting $M = 25,600$ prompts corresponding to a selected task $f$. Next, the first principal component of the column space of these datasets is extracted through SVD. We then report the KL divergence between the KDE-estimated distributions of these components.

## B.6. Computational Resources

The computational experiments were conducted using a high-performance system with an AMD Ryzen Threadripper PRO 3995WX 64-Cores processor, featuring 128 CPU cores with a base frequency of 3.31 GHz and a boost up to 4.31 GHz. The system had 515 GB of RAM. For GPU acceleration, two NVIDIA RTX A6000 GPUs were employed, each with 49,140 MB of memory and a peak power usage of 300W. This setup provided the necessary computational power to efficiently run the transformer models used in our research.

# C. Complementary Results

In this section, we present the complete set of results that could not be accommodated in the main body due to lack of space. Specifically, it includes comprehensive evaluations of the synthetic data, results under distributional shifts, outcomes for the linguistic tasks, and histograms for the ablation studies.

## C.1. Complete Set of Evaluations on Synthetic Data

In addition to the main text, we present results where the LTV is trained with $T_{\mathbf{v}} = T_{\text{train}}$, $T_{\mathbf{v}} = T_{\text{train}} + 1$, and $T_{\mathbf{v}} = T_{\text{max}}$. As expected, we do not observe improved performance for $T_{\mathbf{v}} = T_{\text{train}}$ since the transformer itself already performs adequately. The LTV trained with the maximum number of examples is included for comparison purposes, although it does not provide insights into generalizability, as it is exposed to all $T_{\text{max}}$ examples.

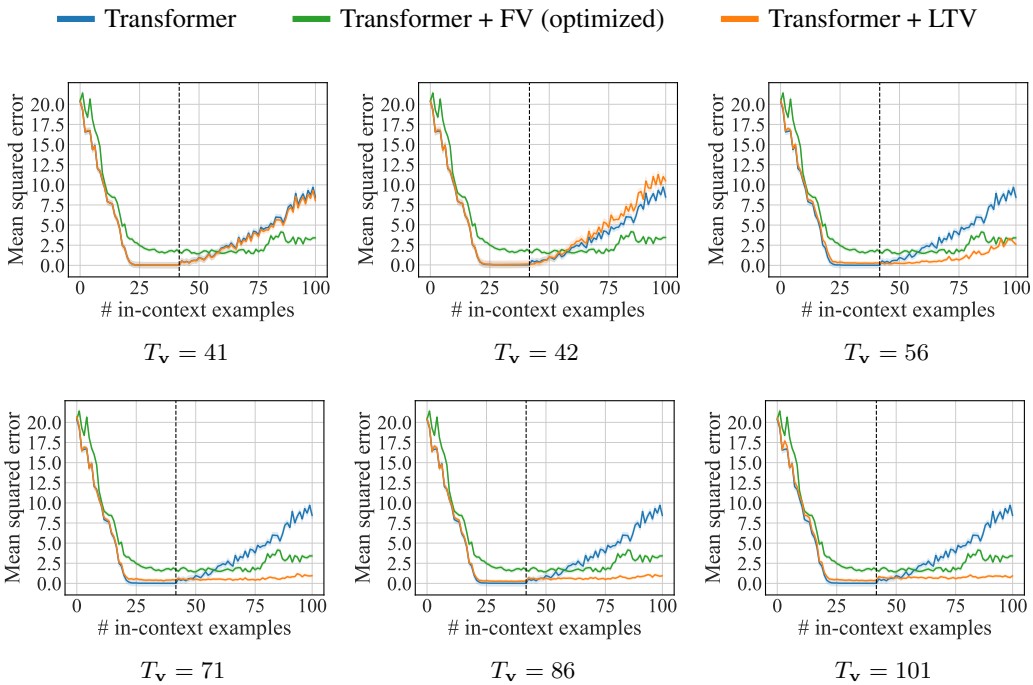

*Figure 6.* Evaluation of the class of linear functions, with the transformer trained with up to $T_{\text{train}} = 41$ examples per prompt. Results are averaged over a batch of 256 randomly selected tasks. The shaded area represents the 95% confidence interval over the sampled prompts. $T_{\mathbf{v}}$ indicates the prompt length used in LTV training.

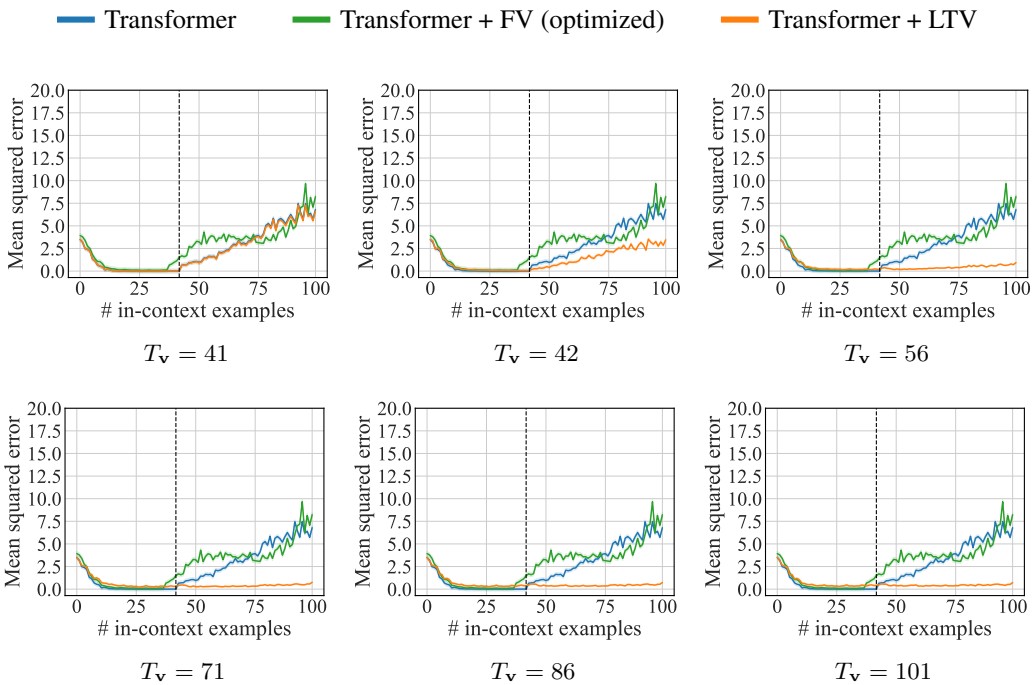

*Figure 7.* Evaluation of the class of sparse linear functions, with the transformer trained with up to $T_{\text{train}} = 41$ examples per prompt. Results are averaged over a batch of 256 randomly selected tasks. The shaded area represents the 95% confidence interval over the sampled prompts. $T_{\mathbf{v}}$ indicates the prompt length used in LTV training.

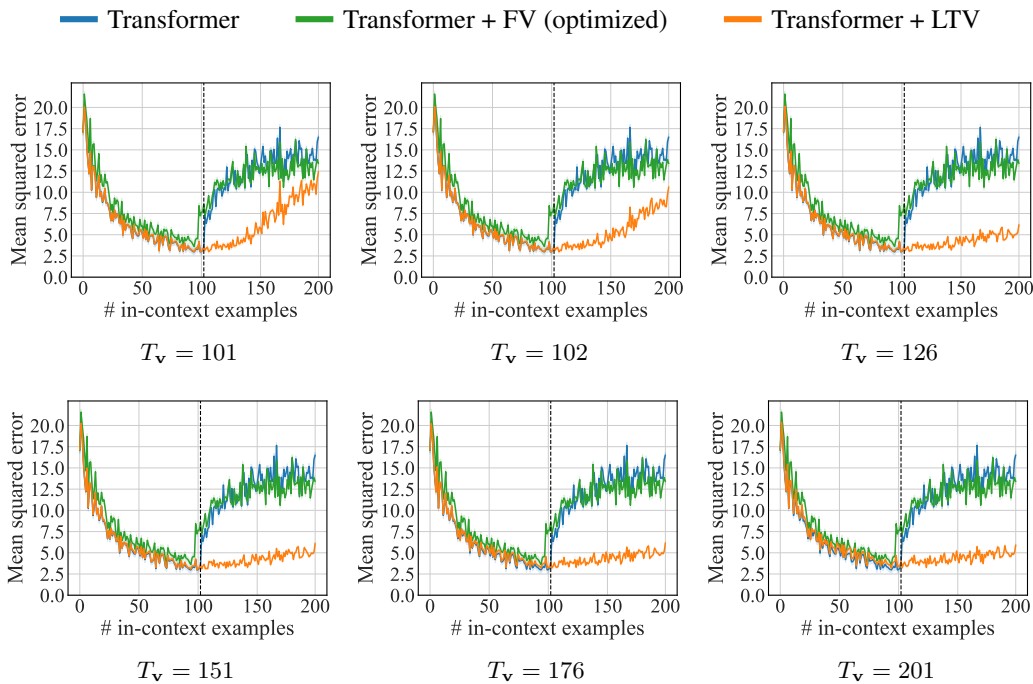

*Figure 8.* Evaluation of the class of 2-layer ReLU neural networks, with the transformer trained with up to $T_{\text{train}} = 101$ examples per prompt. Results are averaged over a batch of 256 randomly selected tasks. The shaded area represents the 95% confidence interval over the sampled prompts. $T_{\mathbf{v}}$ indicates the prompt length used in LTV training.

## C.2. Distributional Shift

We identify two scenarios from (Garg et al., 2023) where the transformer model's performance notably degrades during ICL inference: noisy linear regression and skewed covariance matrix.

**Noisy linear regression**  Noise is added to the output of each example in the form of a standard Gaussian distribution. Specifically, the $i$-th output is defined as $w^\top x_i + \epsilon_i$, where $\epsilon_i \sim \mathcal{N}(0, 1)$. While the transformer and LTV are trained on standard linear regression, the data during the ICL inference phase is modified by this additive noise. We observe that while the performance of FV considerably degrades, LTV is only slightly perturbed. Specifically, LTV requires training on more examples to maintain the same performance compared to the noise-free setting. For instance, the performance of LTV at $T_{\mathbf{v}} = 71$ matches what is achieved in a noise-free environment at around $T_{\mathbf{v}} = 56$.

**Skewed covariance**  The inputs for the prompts are sampled from a zero-mean skewed Gaussian distribution: $x \sim \mathcal{N}(0, \tilde{\Sigma})$, where the eigenbasis of the skewed covariance matrix $\tilde{\Sigma}$ is chosen uniformly at random. Each $i$-th eigenvalue of this matrix is proportional to $1/i^2$. The results we obtain align with the expectations based on (Garg et al., 2023), with the error curves are more unstable and oscillate more than in the isotropic Gaussian case. While LTV also shows some vulnerability to this instability, it still maintains a low mean error and delivers substantial performance enhancement.

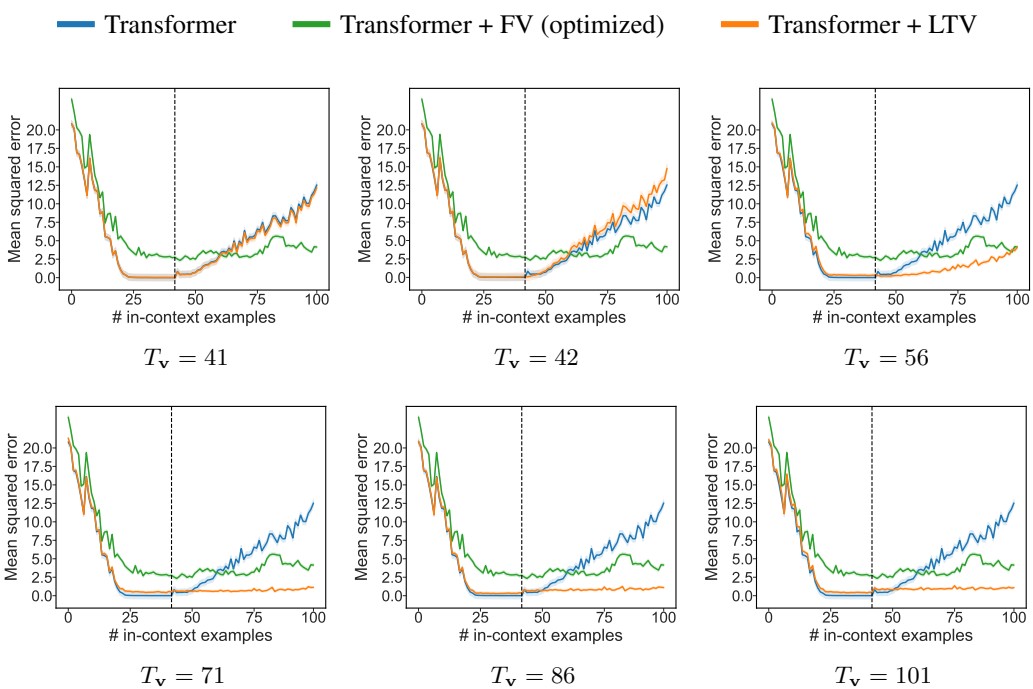

*Figure 9.* Evaluation of the class of linear functions on prompts with noisy labels, with the transformer trained with up to $T_{\text{train}} = 41$ examples per prompt. Results are averaged over a batch of 256 randomly selected tasks. The shaded area represents the 95% confidence interval over the sampled prompts. $T_{\mathbf{v}}$ indicates the prompt length used in LTV training.

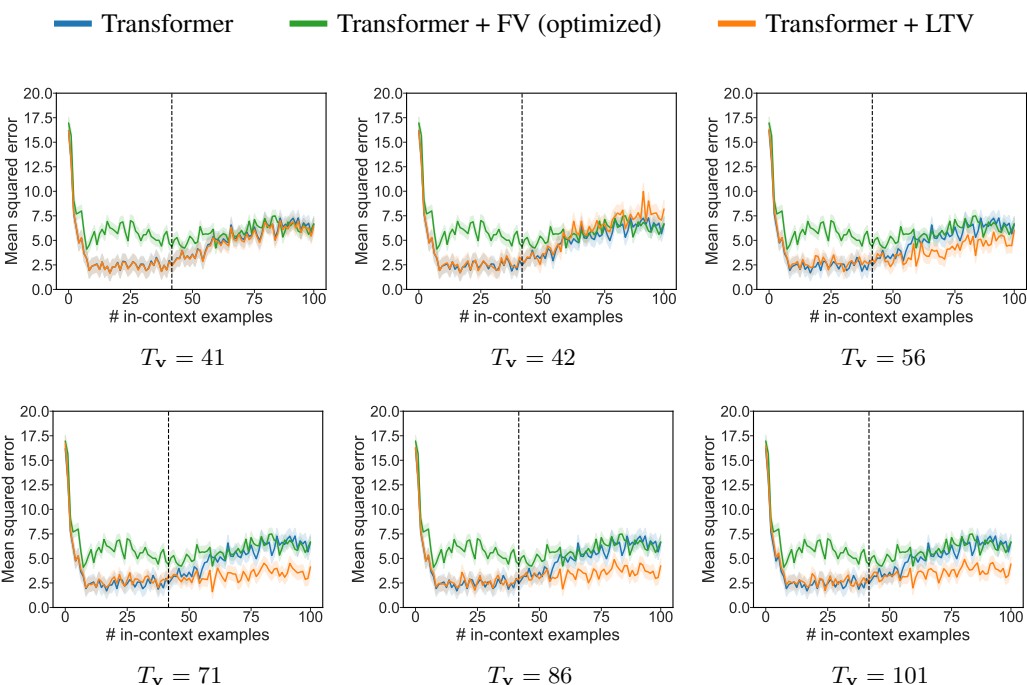

*Figure 10.* Evaluation of the class of linear functions under skewed covariance, with the transformer trained with up to $T_{\text{train}} = 41$ examples per prompt. Results are averaged over a batch of 256 randomly selected tasks. The shaded area represents the 95% confidence interval over the sampled prompts. $T_{\mathbf{v}}$ indicates the prompt length used in LTV training.

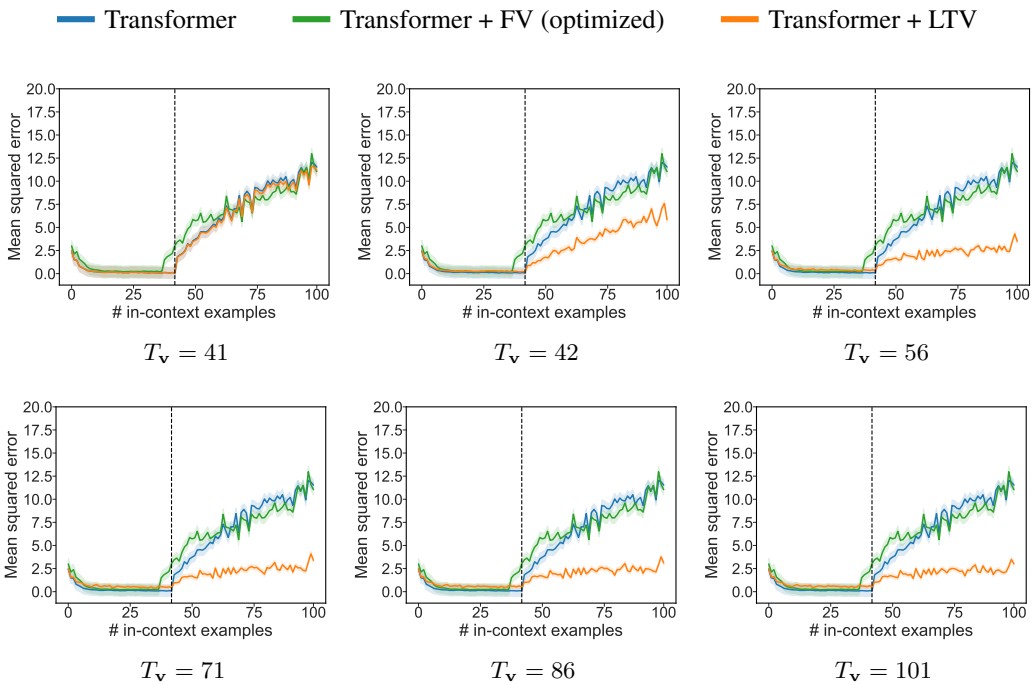

*Figure 11.* Evaluation of the class of sparse linear functions under skewed covariance, with the transformer trained with up to $T_{\text{train}} = 41$ examples per prompt. Results are averaged over a batch of 256 randomly selected tasks. The shaded area represents the 95% confidence interval over the sampled prompts. $T_{\mathbf{v}}$ indicates the prompt length used in LTV training.

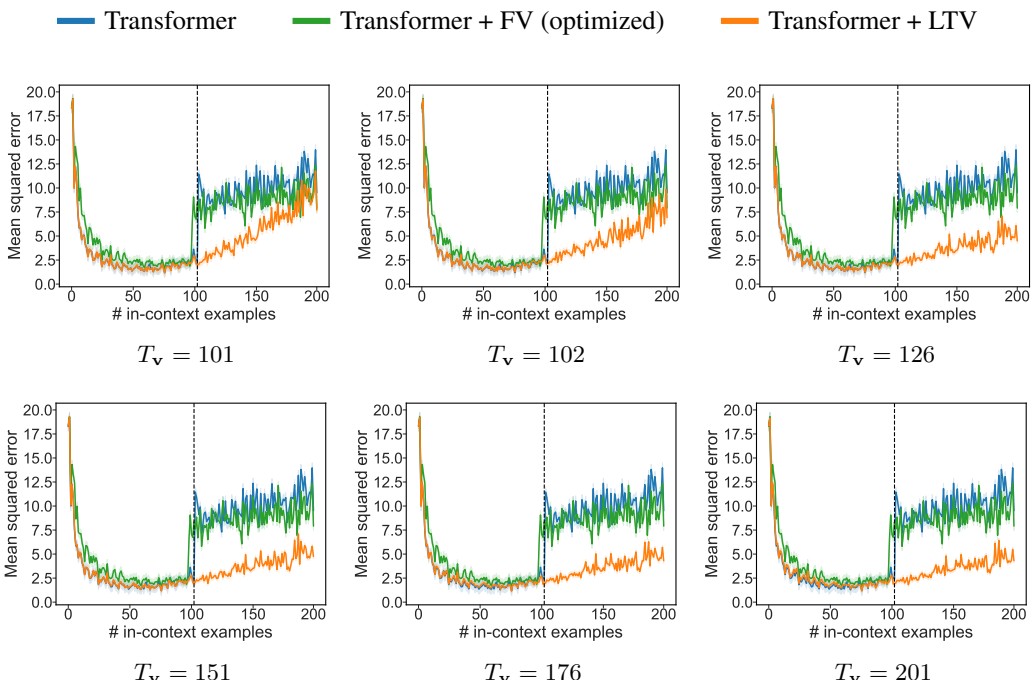

*Figure 12.* Evaluation of the class of 2-layer ReLU neural networks under skewed covariance, with the transformer trained with up to $T_{\text{train}} = 101$ examples per prompt. Results are averaged over a batch of 256 randomly selected tasks. The shaded area represents the 95% confidence interval over the sampled prompts. $T_{\mathbf{v}}$ indicates the prompt length used in LTV training.

### C.3. Additional Results for Language Tasks

For "filtered" accuracy scores, which exclude queries that none of the models predict correctly, the performance differences align with those observed in the unfiltered results. LTV frequently correctly predicts samples that other models do not, a distinction that is particularly evident in zero-shot performance on synonym tasks. LTV achieves an outstanding accuracy score of 0.957, significantly outperforming other models, including the mixed LTV. However, in simpler scenarios, such as few-shot antonym predictions, the performance of mixed LTV and FV is very close to that of the task-specific LTV. This similarity suggests that these configurations generally succeed on the same queries.

The recorded cross-entropy losses correspond to the accuracy scores as we train LTVs specifically to minimize this loss. However, perplexities do not always align with these accuracies. For instance, in the unfiltered zero-shot perplexity scores, FV achieves lower perplexity despite a higher loss. Since perplexity measures the model's uncertainty in its predictions, this discrepancy suggests that while LTV often predicts the correct query, the confidence in its predictions—indicated by the probability assigned to the predicted tokens—can be lower compared to FV. In contrast, FV's predictions are marked by higher probability, suggesting that while LTV is optimized to improve accuracy, it may not necessarily reduce uncertainty as effectively.

*Table 3.* Filtered accuracy scores for few-shot (5-shot) and zero-shot predictions, averaged across 256 random seeds. $\pm$ denotes the 95% confidence interval for the trials. The term "mixed" indicates the LTV weights trained on a joint dataset containing samples from both tasks. The highest accuracy is marked with **boldface** and highlighted.

| Model | Antonym | | Synonym | |
|---|---|---|---|---|
| | Few-shot | Zero-shot | Few-shot | Zero-shot |
| Transformer | $0.457 \pm 0.06$ | $0.027 \pm 0.02$ | $0.102 \pm 0.04$ | $0.035 \pm 0.02$ |
| Transformer + FV | $0.816 \pm 0.05$ | $0.645 \pm 0.06$ | $0.336 \pm 0.06$ | $0.121 \pm 0.04$ |
| Transformer + LTV (mixed) | $0.887 \pm 0.04$ | $0.543 \pm 0.06$ | $0.711 \pm 0.06$ | $0.258 \pm 0.05$ |
| Transformer + LTV | $\mathbf{0.930 \pm 0.03}$ | $\mathbf{0.938 \pm 0.03}$ | $\mathbf{0.926 \pm 0.03}$ | $\mathbf{0.957 \pm 0.03}$ |

*Table 4.* Few-shot perplexity scores and cross-entropy losses corresponding to the results reported in Tables 1 and 3, averaged across 256 random seeds. $\pm$ denotes the 95% confidence interval for the trials. The term "mixed" indicates the LTV weights trained on a joint dataset containing samples from both tasks. The lowest loss and perplexity are marked with **boldface** and highlighted.

| | Model | Antonym | | Synonym | |
|---|---|---|---|---|---|
| | | Perplexity | Loss | Perplexity | Loss |
| Filtered | Transformer | $516.720 \pm 278.41$ | $4.178 \pm 0.25$ | $744.817 \pm 141.77$ | $5.812 \pm 0.16$ |
| | Transformer + FV | $16.541 \pm 7.14$ | $1.860 \pm 0.14$ | $83.423 \pm 14.21$ | $3.964 \pm 0.11$ |
| | Transformer + LTV (mixed) | $2.867 \pm 0.45$ | $0.703 \pm 0.09$ | $5.798 \pm 0.66$ | $1.497 \pm 0.08$ |
| | Transformer + LTV | $\mathbf{2.636 \pm 0.34}$ | $\mathbf{0.685 \pm 0.08}$ | $\mathbf{2.856 \pm 0.39}$ | $\mathbf{0.850 \pm 0.07}$ |
| Unfiltered | Transformer | $1147.430 \pm 598.90$ | $4.786 \pm 0.27$ | $1423.851 \pm 325.89$ | $6.367 \pm 0.17$ |
| | Transformer + FV | $\mathbf{214.981 \pm 186.03}$ | $2.701 \pm 0.23$ | $203.614 \pm 54.01$ | $4.553 \pm 0.14$ |
| | Transformer + LTV (mixed) | $914.691 \pm 1473.71$ | $1.643 \pm 0.24$ | $\mathbf{61.062 \pm 42.07}$ | $2.431 \pm 0.17$ |
| | Transformer + LTV | $852.217 \pm 1514.17$ | $\mathbf{1.625 \pm 0.23}$ | $317.876 \pm 378.42$ | $\mathbf{2.097 \pm 0.21}$ |

*Table 5.* Zero-shot perplexity scores and cross-entropy losses corresponding to the results reported in Tables 1 and 3, averaged across 256 random seeds. $\pm$ denotes the 95% confidence interval for the trials. The term "mixed" indicates the LTV weights trained on a joint dataset containing samples from both tasks. The lowest loss and perplexity are marked with **boldface** and highlighted.

| | Model | Antonym | | Synonym | |
|---|---|---|---|---|---|
| | | Perplexity | Loss | Perplexity | Loss |
| Filtered | Transformer | $500.037 \pm 210.09$ | $4.962 \pm 0.18$ | $552.809 \pm 144.96$ | $5.210 \pm 0.18$ |
| | Transformer + FV | $15.580 \pm 3.52$ | $2.010 \pm 0.13$ | $53.688 \pm 18.70$ | $3.168 \pm 0.14$ |
| | Transformer + LTV (mixed) | $21.237 \pm 8.21$ | $1.682 \pm 0.17$ | $26.866 \pm 9.75$ | $2.329 \pm 0.15$ |
| | Transformer + LTV | $\mathbf{3.685 \pm 1.84}$ | $\mathbf{0.775 \pm 0.09}$ | $\mathbf{5.071 \pm 3.22}$ | $\mathbf{0.984 \pm 0.08}$ |
| Unfiltered | Transformer | $17394.220 \pm 20700.69$ | $6.298 \pm 0.28$ | $5308.716 \pm 2090.15$ | $6.675 \pm 0.24$ |
| | Transformer + FV | $5477.572 \pm 8715.50$ | $3.503 \pm 0.29$ | $\mathbf{604.556 \pm 276.66}$ | $4.645 \pm 0.21$ |
| | Transformer + LTV (mixed) | $29207.165 \pm 41862.46$ | $3.647 \pm 0.34$ | $5270.422 \pm 4532.88$ | $4.623 \pm 0.30$ |
| | Transformer + LTV | $\mathbf{3534.645 \pm 3968.68}$ | $\mathbf{2.496 \pm 0.32}$ | $4392.303 \pm 5894.20$ | $\mathbf{3.301 \pm 0.28}$ |

### C.4. Histograms for Ablation Studies

The histograms effectively illustrate the (unnormalized) probability density functions of the last hidden states. The plots correspond well with the computed KL divergence scores reported in Table 2. As the KL divergence values decrease, the histograms show greater alignment. Specifically, as the training prompt length for the LTV configurations increases, their density profiles become narrower, more closely resembling the shape of the vanilla transformer's distribution at $T_{\text{train}}$. From a different perspective, this visual alignment supports our hypothesis once again: An optimized LTV with sufficiently long prompts performs near-optimally, as it effectively maintains the last hidden state distribution close to that of the model performing under $T = T_{\text{train}}$.

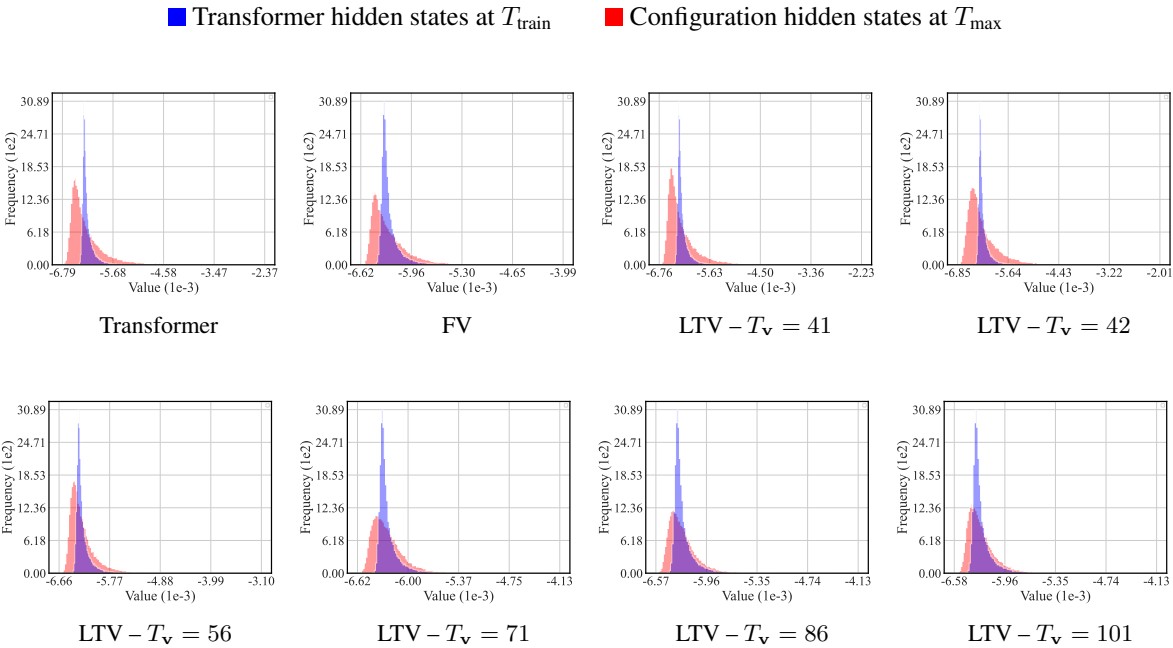

*Figure 13.* Histograms for the empirical distribution of the last hidden states of the vanilla transformer collected at $T_{\text{train}}$ and the tested configuration at the maximum length $T_{\text{max}} = 101$ under linear functions. These histograms are generated using the dataset of 25,600 samples and correspond to the KL divergence scores reported in Table 2.

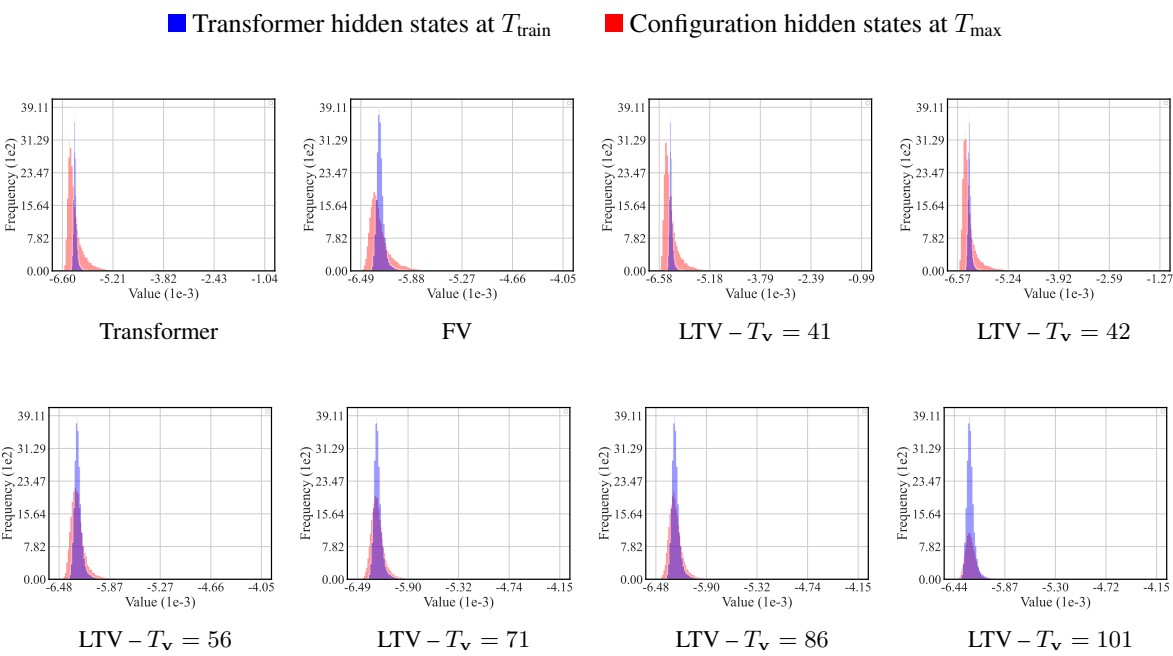

*Figure 14.* Histograms for the empirical distribution of the last hidden states of the vanilla transformer collected at $T_{\text{train}}$ and the tested configuration at the maximum length $T_{\text{max}} = 101$ under sparse linear functions. These histograms are generated using the dataset of 25,600 samples and correspond to the KL divergence scores reported in Table 2.

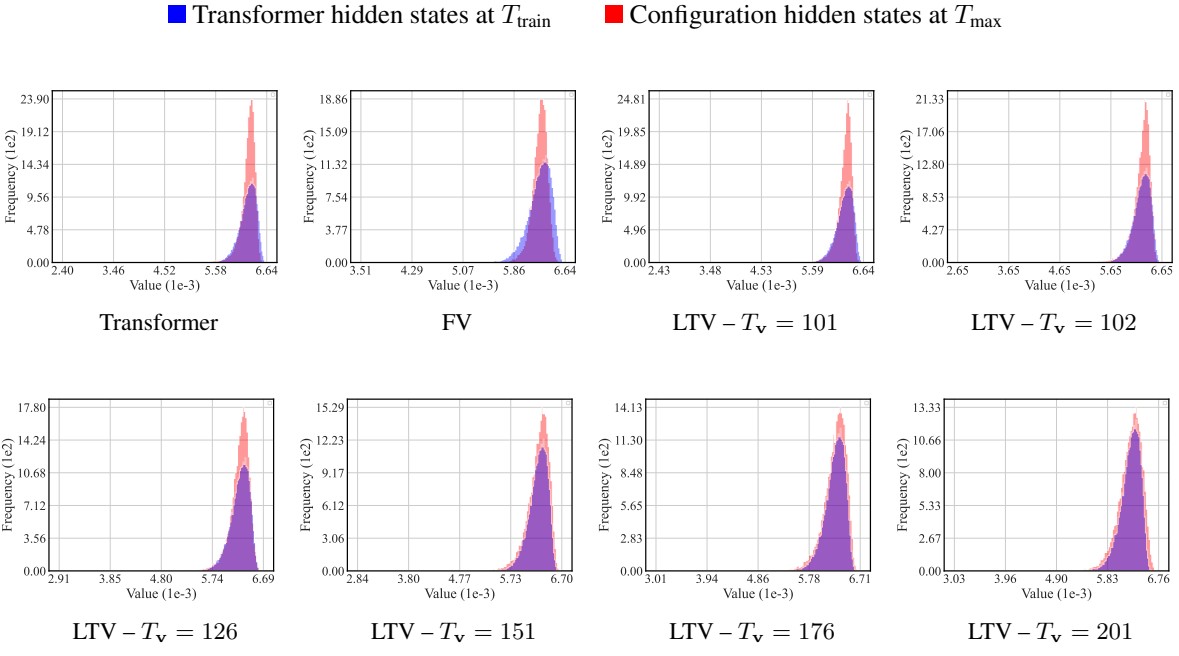

*Figure 15.* Histograms for the empirical distribution of the last hidden states of the vanilla transformer collected at $T_{\text{train}}$ and the tested configuration at the maximum length $T_{\text{max}} = 201$ under 2-layer ReLU neural networks. These histograms are generated using the dataset of 25,600 samples and correspond to the KL divergence scores reported in Table 2.

