# OpenReview forum: "Learning Task Representations from In-Context Learning"
_ICML.cc/2024/Workshop/ICL — ICML 2024 Workshop ICL Poster_

### Official Review · Reviewer_2kHi · 2024-06-03
**Interesting findings based on previous results**

**Rating:** 2
**Fit:** 3
**Confidence:** 2

**Workshop Review:**

This paper presents an extension of function vectors (FVs) proposed in a previous work [1] to improve its effectiveness in deriving task-specific information in in-context learning (ICL). The paper proposes to add a learnable weight to the components of FV (i.e., the attention heads' activations) to "learn" the task representations. The problem formulation and technical details are presented clearly. The authors demonstrate the effectiveness of their method by comparing the loss curve (in terms of MSE) with the original FV and comparing the accuracy of ICL on the synonym and antonym datasets.

**Pros**
- The topic of understanding the internal representation of ICL is of great interest to the community.
- The proposed method is intuitive and effective.

**Cons**
- While the proposed method has been shown to result in better accuracy empirically, adding learned weights deteriorates the interpretability of the FV. It would be interesting to include an analysis of the learned weights. Some examples of questions that I am keen to know the answer for: Which attention heads tend to have greater learned weight? Is there any pattern in the distribution of the learned weight?

[1] Todd, E., Li, M., Sharma, A. S., Mueller, A., Wallace, B. C., and Bau, D. LLMs represent contextual tasks as compact function vectors. In The Twelfth International Conference on Learning Representations, 2024.

**Reason For Not Giving Higher Score:**

The technique proposed is built upon a previous finding for improving the original method.  The originality of the proposed idea is thus limited.

**Reason For Not Giving Lower Score:**

While there is certainly room for improvement in terms of the completeness of the research results, the paper has presented an interesting angle to improve a previous effort in finding a task representation of ICL with empirical evidence. I believe this work fits the workshop clearly and can potentially inspire future research in the area.

---

> ### Author Response · Authors · 2024-07-25
> **Thank you for your comments**
>
> We would like to thank the reviewer for their insightful comments. We would like to emphasize that our findings indicate the learned attention weights are not distributed in an interpretable manner. Specifically, their magnitudes typically lie within the interval of [-3, 3]. However, the distribution varied across different tasks and training durations of the LTV, displaying no consistent pattern. Consequently, we did not identify any specific trends.
>
> In response to the reviewer's comment, we have included a brief paragraph to Section 5.3 (Ablation Studies) to highlight this observation:
>
> > **Learned LTV weights do not exhibit an interpretable pattern.** We observed that the learned attention weights, with magnitudes generally between [-3, 3], showed no consistent distribution pattern across different tasks, training durations of the LTV, or those found by Todd et al. This suggests that LTV weights might capture subtle nuances and be very sensitive to small changes in the training setup, such as the LTV training length.

---

### Official Review · Reviewer_awq2 · 2024-06-04
**Adapting Task Vectors for Synthetic Numeric Tasks**

**Rating:** 2
**Fit:** 3
**Confidence:** 3

**Workshop Review:**

**Summary: **
* This paper finds transformers trained on synthetic numeric tasks fail to generalize when prompts are extended past their training context length. They also find that previous task-vector methods supposed to inject "task information" don't remedy the situation very well. They propose a learnable task vector (LTV) which evolves via several modifications to previous a task vector formulation which is able to maintain good task performance even in long contexts where the original model struggles.

**Clarity:**
* The presentation of the LTV method itself was fairly clear and easy to follow - learn a weighting on the  average attention head outputs that are added back to the residual stream.
* The main thing that was unclear to me was the reason for why the model's performance suddenly gets worse on synthetic number tasks when going beyond the trained length, and also for why you'd need more shots when the task can be done with less. For example, if the model can do the task with 40 shots (error seemed low), why do we need it to work with 80 shots?
* I think the motivation for the method/problems studied could use some more clarification.

**Correctness:**
* The description of the method seems sound, and the analysis presented seems reasonable. It does look like the LTV method is able to overcome the shortcomings of the training-length for the synthetic tasks and works well for the language tasks tested.
* The analysis is limited to a small set of tasks - 2 linguistic and 3 synthetic numeric tasks. So for ICL-use in general it is unclear how well this method/results will hold.

**Novelty:**
* Understanding the limitations of ICL is an interesting area, and testing task vectors in non-language settings seems novel. While learning a weighted mask over many components is not necessarily novel by itself, it sounds like great care has been taken to study different tweaks that are important for having activation-based task vectors work reliably, and that is an interesting result/research to share.
* Understanding how ICL tasks are represented/commuicated in transformers is of interest in the community

**Reason For Not Giving Higher Score:**

* There is no investigation into what is learned by the optimization process. Do the learned weights match the heads that Todd et al's formulation uses, or is it very different depending on the task? Does the weighting provide any insight into the LTV construction? This seems like a key claim, but is not explored from what I could see.

* The evaluation of LTVs is limited to 2 simple linguistic tasks (antonyms, synonyms), and 3 synthetic numeric tasks. More tasks should be evaluated to understand the extent to which LTVs generalize.

* The paper's related work section (Section 2 & Appendix A) seems highly similar to that found in Todd et al's paper, which this paper builds on (i.e. mostly the same papers cited, exact same order of citing, etc., which is atypical) though the final paragraph of Appendix A is new.

* One other concern is regarding the sharp performance dip of the model when given prompts that are longer than what it is trained with. While the authors note this is one of the main motivations of their study, it is very loosely connected in my opinion to ICL task vectors, and they don't explore it in detail at all. I think more study should be done to understand what is causing this shift in behavior.
It seems like the model's performance for sequence lengths less than or equal to what it is trained with is pretty good on the synthetic tasks, and so it doesn't seem likely the model suddenly cannot "figure out" what the task is anymore. I think a better understanding of why this is happening would strengthen the paper. One possibility is that is could be a limitation of the architectural design choices. For example, it was mentioned to be a GPT2 model -- was it trained with absolute position embeddings? In this case, extending beyond training length can sometimes cause problems, and then this phenomena might not be as surprising, though it is still interesting that LTV works in such a case.

**Reason For Not Giving Lower Score:**

* The presentation of the LTV method was clear and straightforward.

* For the cases they tested, the LTV method's empirical results show it alleviates the problem identified by the authors of a performance drop when prompt length becomes longer than what it was trained with.

* The study of different elements of how to use activation-based task-vectors seems valuable, and this paper seems to have done a careful study into what sorts of things do and don't work (both synthetic and language based.)

* They study task vectors for non-linguistic settings, which is an interesting and novel application - studying how previous work can be adapted to a new setting.

---

> ### Author Response · Authors · 2024-07-25
> **Thank you for your constructive feedback**
>
> We thank the reviewer for their constructive comments. Here is our point-by-point response to the reviewer's feedback.
>
> - **Interpretation of the learned weights:** Our work did not indicate any specific pattern in the learned LTV weights, showing neither a consistent distribution nor matching Todd et al.'s formulation. To reflect this in the paper, we have included a concise paragraph in Section 5.3 (Ablation Studies):
> > **Learned LTV weights do not exhibit an interpretable pattern.** We observed that the learned attention weights, with magnitudes generally between [-3, 3], showed no consistent distribution pattern across different tasks, training durations of the LTV, or those found by Todd et al. This suggests that LTV weights might capture subtle nuances and be very sensitive to small changes in the training setup, such as the LTV training length.
>
> - **More tasks:** We also conducted experiments with the decision tree task, where the input-output relationship is defined by a tree of depth 4. However, we did not observe any performance deterioration for prompts longer than the pre-training length, as highlighted at the beginning of Section 5, under the paragraph titled Tasks. Additional benchmark tasks are presented in Appendix C.2, showing results under distributional shifts. We investigate performance in two more benchmarks: noisy linear regression and data sampled from a zero-mean Gaussian with skewed covariance.
>
> - **Identification of the length generalization problem and connections to positional encodings:** The model is indeed a GPT-2, but we used the modern implementation from [Hugging Face](https://github.com/huggingface/transformers/blob/main/src/transformers/models/gpt2/modeling_gpt2.py), which uses learnable positional encodings (PE). For example, in the linear regression task, the model is initialized to accommodate sequences of 101 examples but sees only up to 41 examples during pre-training. Hence, we attribute the poor ICL performance to the fact that approximately 60% of the PE weights remain untrained. The LTV, on the other hand, can remedy this by interpolating to longer sequences.

---

### Meta-Review · Area_Chair_BCVt · 2024-06-16

**Recommendation:** 2

**Metareview:**

This paper addresses a significant challenge in the field of ICL by proposing a novel approach to enhance the generalization capabilities of transformers on synthetic numeric tasks. The clarity of the method's presentation is commendable, particularly the explanation of learning a weighting on the average attention head outputs. However, the motivation behind the sudden performance drop in longer contexts and the necessity for more shots in certain tasks require further clarification. The correctness of the method is well-supported by the analysis, though the scope of tasks tested is somewhat limited. Including an analysis of the learned weights could further enhance the paper's impact. However, the paper makes a valuable contribution to the field and is recommended for acceptance.

---

> ### Author Response · Authors · 2024-07-25
> **Thank you for your feedback**
>
> We would like to thank you for your thoughtful and constructive feedback. We appreciate the recognition of our work's significance and the clarity of our method's presentation. We have addressed the points raised by the reviewers in the revised paper.

---

### Decision · Program_Chairs · 2024-06-17

**Decision:**

Accept (Poster)

**Comment:**

**Accept with minor revision**: We would like to accept this work as it explores relevant topic in ICL. However, we note that the scope of the claims in this work seems not appropriate with regard to the assumptions made and some relevant work is missed where different experimental evidence exists with modified conclusion. Particularly length generalization has been observed on synthetic task with transformers, e.g. in "What Algorithms can Transformers Learn? A Study in Length Generalization", "TabPFN: A Transformer That Solves Small Tabular Classification Problems in a Second" and the particular architectural settings have e.g. been studied in "Statistical Foundations of Prior-Data Fitted Networks". We ask the authors to clarify in which conditions claims are made and discuss related approaches and findings.

---

> ### Author Response · Authors · 2024-07-25
> **Response to paper decision**
>
> We would like to thank the meta-reviewer for their valuable feedback and for considering our work for acceptance. We appreciate the references to related work. However, we believe that the mentioned studies are not directly relevant to our proposed method as we examine the special case of in-context learning.